



# Common features of iodate to iodide reduction amongst a diverse range of marine phytoplankton

Helmke Hepach[1*+], Claire Hughes[1+], Karen Hogg[2], Susannah Collings[1] and Rosie Chance[3]

[1]Department of Environment and Geography, University of York, York, UK
[2]Department of Biology, University of York, York, UK
[3]Wolfson Atmospheric Chemistry Laboratory (WACL), University of York, York, UK

*now at GEOMAR Helmholtz Centre for Ocean Research Kiel, RD2, Biological oceanography, Kiel, Germany

+authors contributed equally to the preparation of this manuscript

*Correspondence to*: Helmke Hepach (hhepach@geomar.de)

**Abstract.** The reaction between ozone and iodide at the sea surface is now known to be an important part of atmospheric ozone cycling, causing ozone deposition and the release of ozone-depleting reactive iodine to the atmosphere. The importance of this reaction is reflected by its inclusion in chemical transport models (CTMs). Such models depend on accurate sea surface iodide fields but measurements are spatially and temporally limited. The ability to predict current and future sea surface iodide fields requires the development of process-based models which in turn require a thorough understanding of the key processes controlling inorganic iodine cycling. The aim of this study was to inform the development of ocean iodine cycling models by exploring if there are common features of iodate to iodide reduction amongst diverse marine phytoplankton. In order to achieve this, rates and patterns of changes in inorganic iodine speciation were determined in 10 phytoplankton cultures grown at ambient iodate concentrations. Where possible these data were analysed alongside results from previous studies. Iodate loss and some iodide production was observed in all cultures studied, confirming that this is a widespread feature amongst marine phytoplankton. We found no significant difference in log-phase, cell-normalised iodide production rates between key phytoplankton groups (diatoms, prymesiophytes including coccolithophores and phaeocystales) suggesting that a Phytoplankton Functional Type (PFT) approach would not be appropriate for building an ocean iodine cycling model. Iodate loss was greater than iodide formation in the majority of the cultures studied, indicating the presence of an as yet unidentified 'missing iodine' fraction. Iodide yield at the end of the experiment was significantly greater in cultures that had reached a later senescence stage. This suggests that models should incorporate a lag between peak phytoplankton biomass and maximum iodide production, and that cell mortality terms in biogeochemical models could be used to parameterize iodide production.

## 1  Introduction

Interest in marine inorganic iodine has increased in recent years due to the realisation that ozone deposition to iodide ($I^-$) at the sea surface plays an important role in ozone cycling and the release of reactive iodine to the troposphere (Carpenter et al., 2013). Once tropospheric ozone reacts with iodide, both hypoiodous acid and molecular iodine are produced in the sea surface microlayer and are then released to the atmosphere (Carpenter et al., 2013; MacDonald et al., 2014). Prados-Roman et al. (2015) suggest that up to 75 % of total iodine oxide found in the marine troposphere may be originating from this reaction. Once in the troposphere reactive iodine takes part in numerous chemical cycles and reactions which impact the $HO_x$- and $NO_x$-cycle and ozone cycling



(Saiz-Lopez et al., 2012). Iodine is also known to be involved in new particle formation (Ehn et al., 2010; O'Dowd et al., 2002; Sellegri et al., 2016). Ozone deposition to iodide at the sea surface is now considered to be
an important component of atmospheric chemistry and is incorporated into large scale Chemical Transport Models (Luhar et al., 2017; Sherwen et al., 2016).

Given the link to atmospheric processes there is now an increased need for accurate maps of global ocean sea surface iodide fields (Carpenter et al., 2013; Helmig et al., 2012). As highlighted by Chance et al. (2014) direct
measurements of iodide in surface seawater are sparse. Hence, for the purpose of estimating large-scale ozone deposition and iodine emissions sea surface iodide concentrations have been estimated as a function of oceanographic variables such as nitrate (Ganzeveld et al., 2009), temperature (Sherwen et al., 2016; Luhar et al., 2017) and chlorophyll $a$ (Helmig et al., 2012; Oh et al., 2008), and more recently using combinations of variables (Sherwen et al., 2019). Improving the ability to predict current and future sea surface iodide fields
requires the development of process-based models and that in turn requires a better understanding of the key processes controlling inorganic iodine cycling in marine systems.

Existing measurements show that total inorganic iodine, mainly consisting of iodide and iodate ($IO_3^-$), is found throughout the oceans at a fairly constant concentration of 450 – 500 nM (e.g. Elderfield and Truesdale 1980;
Truesdale et al. 2000) but that the ratio of iodide and iodate has considerable spatial variability. In general, iodate occurs at higher concentrations in seawater than iodide throughout most of the water column but elevated iodide concentrations are found towards the surface (Chance et al., 2014). Highest sea surface iodide concentrations (greater than 100 nM) occur in low latitude waters, while latitudes greater than about 40° N or S are characterised with concentrations of less than 50 nM (Chance et al., 2014). Chance et al. (2014) also reported
that a number of studies have observed a decrease in the proportion of dissolved iodine present as iodate in coastal waters. Given that ozone deposition is proportional to the concentration of iodide at the sea surface (Carpenter et al., 2013), this spatial variability will have a major impact on atmospheric ozone cycling.

Iodide concentrations in seawater are thought to be predominantly controlled by loss due to oxidation to iodate,
production due to iodate reduction and physical mixing (reviewed by Chance et al., 2014). Estimates of the lifetime of iodide due to oxidation range between less than six months and 40 years (Campos et al., 1996; Edwards and Truesdale, 1997; Tsunogai, 1971). Reported abiotic rates are too slow to explain the shorter lifetimes so biogenic iodide oxidation driven by phytoplankton (Bluhm et al., 2010) or bacteria (Amachi, 2008; Fuse et al., 2003; Zic et al., 2013) has been suggested but there remains great uncertainty surrounding this
process (Truesdale, 2007). Studies have revealed that both photochemical (Miyake and Tsunogai, 1963; Spokes and Liss, 1996) and biological processes (Bluhm et al., 2010; Chance et al., 2009; Chance et al., 2007; Kupper et al., 1997) are involved in iodate to iodide reduction. Calculations suggest that the photochemical reduction of iodate is too slow to be of significance (Truesdale, 2007). Hence we need a greater understanding of biological iodine cycling in order to develop ocean cycling models that can inform studies of ozone deposition to seawater
and sea-air iodine emissions.

The reduction of iodate to iodide has been observed in unialgal cultures representative of a wide range of different phytoplankton groups including diatoms, prymnesiophytes and cyanobacteria (Bluhm et al., 2010;



Chance et al., 2007; Moisan et al., 1994; van Bergeijk et al., 2016; Waite and Truesdale, 2003; Wong et al., 2002). Whilst this demonstrates that the process is widespread amongst marine primary producers, the patterns and rates observed are hugely variable. To date, the highest rates of iodate to iodide conversion observed at ambient iodate concentrations (300 nM) have been seen in the cold water diatom *Nitzschia* sp. (CCMP 580), which has been found to mediate production at 123 amol $I^-$ cell$^{-1}$ d$^{-1}$ (Chance et al., 2007) but there is currently insufficient coverage to establish which (if any) algal groups dominate. There is some evidence to suggest that iodate to iodide reduction in marine phytoplankton is to some extent controlled by environmental conditions (e.g. iodate concentration, van Bergeijk et al., 2016) but to date no systematic study has been undertaken to establish the dominant controls. Hence we are unable at present to establish if there are common features of iodate to iodide reduction amongst diverse marine phytoplankton or identify the environmental drivers.

The exact processes involved in iodate to iodide reduction and its metabolic function (if any) in marine phytoplankton remains uncertain but suggestions include links with nitrate reductase (Tsunogai and Sase, 1969) and senescence (Bluhm et al., 2010). Indications for the link with nitrate reductase come from correlations between iodide concentration and nitrate reductase activity in the field (Wong and Hung, 2001) and from laboratory studies with enzyme extracts (Hung et al., 2005; Tsunogai and Sase, 1969). There is, however, also evidence from culture studies, which suggests that nitrate reductase is not involved in iodate to iodide conversion. For instance, Waite and Truesdale (2003) deactivated the nitrate reductase enzyme in a haptophyte species which was still able to reduce iodate to iodide and Bluhm et al. (2010) did not see a link between iodide production with nitrate limitation in their monoculture studies. They instead suggested a link of iodide production with senescence mediated by reduced sulfur leaked from lysing cells. To date, this was the only study that tied iodide release to a specific growth phase of microalgae. Studies (Kupper et al., 1997, 2008) have suggested that iodate to iodide reduction in marine macroalgae is linked to light-induced oxidative stress. Whilst iodide has been shown to control oxidative stress in microalgae (Javier et al., 2018), a link between iodate reduction and light-induced oxidative stress has yet to be demonstrated in this group of organisms. A better understanding of the purpose and mechanism of iodate to iodide reduction in marine phytoplankton would help with the development of process-based models of inorganic iodine cycling in the oceans.

The aim of this study was to establish if there are any general trends of iodate to iodide reduction across a diverse range of phytoplankton species to assist with future predictive model development. We studied growth-stage-specific and overall changes in iodate to iodide conversion at ambient iodate concentrations (~300 nM) in 10 polar, temperate and tropical phytoplankton species from three microalgal groups including diatoms, prymnesiophytes, and cyanobacteria. Where possible, we have compiled the rates observed with those from the literature to provide an overall view of patterns of iodate to iodide conversion across the marine phytoplankton cultures studied to date.

## 2    Materials and Methods

### 2.1  Phytoplankton strains

The 10 phytoplankton strains (Fig. 1) used in this study were obtained from the Roscoff Culture Collection (RCC) and the National Center for Marine Algae and Microbiota Bigelow (CCMP). The strains include the



diatoms *Chaetoceros gelidus* (RCC 4512), *Chaetoceros* sp. (RCC 4208), and *Chaetoceros* sp. (CCMP 1690); the prymnesiophytes *Emiliana huxleyi* (RCC 1210, coccolithophore), *Emiliana huxleyi* (RCC 4560,

coccolithophore), *Calcidiscus leptoporus* (RCC 1164, coccolithophore), *Geophyrocapsa oceanica* (RCC 1318, coccolithophore), *Phaeocystis antarctica* (RCC 4024, phaeocystales), and *Phaeocystis* sp. (RCC 1725, phaeocystales); and, the cyanobacterium *Synechococcus* sp. (RCC 2366). Where we studied strains of the same genus or species they were from different climate zones. All cultures were non-axenic but checked for bacterial growth in the beginning and the end of the experiments using flow cytometry (see *section 2.2.2*).

**2.2 Experimental set-up**

Each strain was grown under the conditions (i.e. temperature, light intensity, media) (Table 1) specified by the culture collection from which they were obtained (Fig. 1). All media were prepared in ESAW – enriched seawater, artificial water (Berges et al., 2001), which was autoclaved before use. Handling of all sterile media and cultures was done in a biosafety cabinet to reduce the risk of bacterial contamination. Each experiment

included triplicate phytoplankton cultures and triplicate media-only controls. The duration of the experiment was dictated by the growth dynamics of the specific strain. Each experiment was carried out until the respective culture reached the senescent phase but due to time constraints cultures were at different stages of senescence when each experiment was terminated.

Experiments were performed in either 2 or 4 L borosilicate glass flasks, which contained 1 or 2 L of medium, respectively. The experimental as well as the control flasks were spiked with iodate at a final concentration between 300 and 400 nM (Tables 1 and 2) reflecting natural concentrations (Chance et al., 2014). Initial Iodide concentrations in the flasks ranged on average between $1.32 \pm 0.23$ nM (*Phaeocystes* sp., RCC 1725) and $20.77 \pm 20.49$ nM (*Chaetoceros gelidus*, RCC 4521). Iodate solutions were prepared in Milli Q water using solid

potassium iodate ($KIO_3$, Fisher Scientific, SLR grade, $>= 99.5$ %), and were autoclaved before being added to the ESAW. At the start of each experiment experimental flasks were inoculated with $15 - 30$ mL of stock culture, depending on the stock culture cell density and volume of the flasks. Flasks were then incubated under red and blue LED lights with a 12:12 h light-dark cycle in a temperature controlled room. Regular (weekly or 2-weekly) sampling was performed for inorganic iodine species ($I^-$, $IO_3^-$), cell counts and *in vivo* chlorophyll

fluorescence readings. Methods used for the determination of these parameters are described in *Sections 2.2.1 to 2.2.2*.

**2.2.1    Determination of iodide and iodate**

Samples for iodide and iodate analyses were gently hand-filtered through a 25 mm GF/F (Whatman) filter and then stored at -20° C until further analysis within 6 months of collection. Our storage tests revealed inorganic

iodine speciation was maintained during this period of storage.

Iodide analysis was performed using cathodic stripping square wave voltammetry as described in Campos (1997) using a Metrohm voltammeter and NOVA software. The sample volume was 12 mL and nitrogen was used as purging gas to remove oxygen. 90 µL of 0.2 % Triton-X 100 was added to the sample before purging to increase

the sensitivity of the method. Quantification of iodide was achieved by performing standard additions. Potassium iodide (KI, Acros organics, extra pure, trace metal basis, 99.995 %) standards were prepared in Milli Q water



with a final concentration of about $1 - 2 \times 10^{-5}$ M. Final concentrations were determined by applying linear regression. Precision of the technique was 5 – 10 % based on repeat measurements of aliquots from the same sample.


Iodate was determined spectrophotometrically according to Truesdale and Spencer (1974) using a Perkin Elmer Lambda 35 UV/Vis spectrometer with a 1 cm quartz cuvette. During analysis 50 µL of 1.5 M sulfamic acid (Fisher scientific, Analytical Reagent grade, $\geq$ 99.9 %) was added to 2.3 mL of sample and absorbance at 350 nm was measured after 1 min. Following this, 150 µL of 0.6 M KI solution was added, mixed and the absorbance at 350 nm read after 2.5 min. Quantification was achieved by performing a standard curve on every measurement day using potassium iodate (see *section 2.2*) in Milli Q water. Final iodate concentrations were then retrieved using the difference of the second reading and the first reading and by linear regression from the standards. Sample precision laid between 5 and 10 % based on regular measurements of triplicates from the same sample. A standard in the measured concentration range was measured every five samples to determine the daily instrumental drift.

### 2.2.2   Biological measurements: in vivo chlorophyll fluorescence and cell counts

*In vivo* chlorophyll fluorescence was measured at every sampling point for each culture- and control replicate. A sample of 5 mL was transferred into a 1 cm cuvette and fluorescence (excitation/emission 460 nm/685 nm) was measured using a Turner Trilogy Designs fluorometer.


Automated cell counts were performed using a Vi-Cell XR (Beckman Coulter). 500 µL of the sample were transferred into a vial, the Vi-Cell takes up the sample, mixes it 1:1 with trypan blue, dead cells take up the dye while live cells do not, and delivers it to a flow cell for camera imaging where differences in grey scale between live and dead cell is determined by the software. 50 images were analysed to determine the cell concentration and viability. The ESAW sample had to be acidified with 5 µL of concentrated hydrochloric acid directly prior injecting into the Vi-Cell due to the alkaline nature of the trypan blue and consequential precipitation of sea salt that complicated the measurement using the Vi-Cell without pre-acidification. Pre-tests with acidification and *in vivo* chlorophyll fluorescence of different algal cultures showed no change in the fluorescence with pre-acidification in this short time period. The precision of these measurements was on average 10 % determined from triplicates from the same culture. No cells were detected in the control treatments. Bacterial contamination was evaluated using flow cytometry. Samples were stained with DAPI ($1\mu g\ mL^{-1}$) for 30 minutes at room temperature and analysed using a CytoFLEX S cytometer (Beckman Coulter) using an event rate of less than events 1,000 per second, at flow rate 60 µL per minute for a minimum of 1 minute. DAPI was excited using the 405 nm laser and emitted photons detected in the wavelength range 450/50 nm. Sterile sample diluent buffer was used to set the detection threshold and a sterile media was used as a negative control.

### 2.3  Calculations and data analysis

Iodide production and iodate incorporation rates were calculated from slopes applying linear regression analysis of iodide and iodate concentration versus time according to Bluhm et al. (2010). Pearson's linear correlation coefficients ($R^2$) were generally larger or equal to 0.7 with the exception of one culture replicate of *Emiliana*





*huxleyi* (RCC 4566), where $R^2$ was 0.5. Iodine production and incorporation rates per cells were normalised to time-averaged cell numbers.

All statistical tests applied in this study were conducted in Matlab® and Sigma Plot Version 13. Datasets that were correlated to each other were first tested for normal distribution using the Liliiefors test. Depending on the outcome of the test, linearity was calculated using the Pearson's linear correlation coefficient (R) or the

Spearman's Rho ($r_s$). The significance level applied here was $p \leq 0.05$. Further statistical tests applied include the t-test, the two-sided Wilcoxon rank sum test, and One-way ANOVAs. The latter was used when the means of more than two datasets were investigated at the same time.

Normal distribution of datasets used in the One-way ANOVAs was tested using the Shapiro-Wilk test. Since most datasets were not normally distributed, we performed Kruskal-Wallis One-way ANOVAs, since they do

neither require normal distribution, nor equal variances. The specific purpose of each test is introduced at relevant points in *Section 3*.

## 3    Results and discussion

The results of the phytoplankton growth curve experiments are summarized in Fig. 2 – 5. It can be seen that a decline in iodate concentrations and increase in iodide was detected in all strains studied. Concentrations of

iodide in the media-only controls (Table 2) were very close to detection limit, and thus the small changes observed are within our measurement error. Additionally any changes in iodate observed in the controls were within the precision of the spectrometric method. This confirms that the observed changes in inorganic iodine in the cultures were biologically mediated. It is apparent from Fig. 2 – 5 that there is variability in the time-series and magnitude of changes in iodate and iodide concentrations between cultures but growth rates, biomass levels

and the growth stage reached also differed between strains. The data are explored further in *Sections 3.1 to 3.3* to identify if any common features or patterns of inorganic iodine speciation change can be identified once these other factors are taken into account.

This study did not set out to identify the mechanism of iodate to iodide conversion in marine phytoplankton but we can say that it is unlikely nitrate reductase (Hung et al., 2005) was the mechanism responsible. It was

postulated that the responsible enzyme switches to iodate once nitrate is depleted (Tsunogai and Sase, 1969). Media used to grow each strain in this study however contains high levels of nitrate (441.0 μM in K/2, Keller et al., 1987; 882.0 μM in f/2, Guillard and Ryther, 1962; and K, Keller et al., 1987; 2.5 μM in SN, Waterbury et al., 1986) so the cultures were not limited in this nutrient. The other proposed mechanism for iodate reduction to iodide involves the release of reduced sulfur during the senescence phase (Bluhm et al., 2010). Our further

analysis in *Sections 3.1 to 3.3* explores the importance of growth stage on changes in inorganic iodine speciation and hence goes some way to explore if the mechanism described in Bluhm et al. (2010) can explain the observed changes.

### 3.1 Logarithmic stage rates of iodate to iodide reduction

#### 3.1.1    Cell-normalised rates

Log-phase, cell-normalised iodide production rates were calculated (Table 1) to assess if normalising to biomass allows any patterns to be identified across phytoplankton strains. Our rates are presented in Table 1 alongside those reported in previous studies for comparison. Rates observed in *Synechococcus* sp. (RCC 2366) are not





discussed further here as it was the only cyanobacterium strain studied. Overall we observed the highest rate of iodide production (95.5 ± 19.5 amol I⁻ cell⁻¹ d⁻¹) in the Prymnesiophyte *Calcidiscus leptoporus* (RCC 1164). The

warm water (20° C) *Phaeocystis* sp. (RCC 1725) also had high rates of change of inorganic iodine speciation (60.9 ± 22.5 amol I⁻ cell⁻¹ d⁻¹) but the cold-water *Phaeocystis antarctica* (RCC 4024) had relatively lower rates (3.5 ± 0.9 amol I⁻ cell⁻¹ d⁻¹). The *Emiliana huxleyi* strains investigated here (RCC 1210, RCC 4560) were both found to drive low rates of change in inorganic iodine speciation (< 2 amol I⁻ cell⁻¹ d⁻¹). Other studies have found rates of iodide production in *Emiliana huxleyi* of 66.3 amol I⁻ cell⁻¹ d⁻¹ (CCMP 373, 300 nM iodate, Chance et

al., 2007) and 9 ± 5 to 11 ± 2 amol I⁻ cell⁻¹ d⁻¹ (CCMP 371, at 5 µM iodate, Bluhm et al., 2010). The only other Prymnesiophyte investigated to date (*Tisochrysis lutea*, CCAP 927/14) has been found to produce iodide at rates of 0.7 amol I⁻ cell⁻¹ d⁻¹ at 500 nM iodate and 195.1 nM iodate at 2.5 mM (van Bergeijk et al., 2016).

For the sake of comparability, we concentrate only on studies that reported iodide production and iodate

consumption rates normalised to phytoplankton cell numbers in the following. Across all studies on iodate to iodide reduction by phytoplankton undertaken to date that also include phytoplankton cell numbers the highest rates of iodide production have been observed in diatoms but this was not the case in our study. The cold-water *Nitzschia* sp. (CCMP 580) has been found to mediate 123 amol I⁻ cell⁻¹ d⁻¹ at 300 nM iodate (Chance et al., 2007). The very high rates of diatom iodate to iodide conversion reported in Table 1 from other studies were

observed when the cultures were presented with super-ambient concentrations of iodate (e.g. *Nitzschia* sp., CCMP 580, 8600 amol I⁻ cell⁻¹ d⁻¹ at 10 µM iodate, Chance et al., 2007; *Pseudo-nitzschia turgiduloides*, 643 ± 179 amol I⁻ cell⁻¹ d⁻¹; *Fragilariopsis kerguelensis* 93 ± 19 amol I⁻ cell⁻¹ d⁻¹ both at 5 µM iodate, Bluhm et al. 2010). Whilst the increased iodate to iodide reduction at the higher levels of iodate is of interest, such rates are unlikely to occur in the natural environment, especially since iodide release rates have been shown to increase

with increasing initial iodate concentrations (e.g. Wong et al., 2002; van Bergeijk et al., 2016). The highest rate of iodide increase we observed amongst the diatoms studied here was in the temperate *Chaetoceros* sp. (RCC 4208; 16.6 ± 2.4 amol I⁻ cell⁻¹ d⁻¹). Relatively lower rates were observed in the other two cold-water and temperate *Chaetoceros* strains (Table 1: *Chaetoceros gelidus*, RCC 4512; *Chaetoceros* sp., CCMP 1690). Similarly low rates have been found in other marine diatoms (e.g. *Phaeodactylum tricornutum*, CCMP 1055/15;

van Bergeijk et al., 2016) and some diatom cultures (e.g. *Thalassiosira pseudonana*, CCMP 1335, Chance et al. 2007) have not been found to mediate iodate to iodide reduction at all.

A similar wide range in iodide production and iodate consumption rates was found for monoculture batch experiments where no cell-normalised rates were presented (Butler et al., 1981; Moisan et al., 1994; Wong et al., 2002; Waite and Truesdale, 2003). For example, while Wong et al. (2002) present high iodide production rates in

monocultures of the green alga *Dunaliella tertiolecta*, Butler et al. (1981) did not see any changes in iodide levels in their experiments with the same species.

Overall, when all cell-normalised iodide production rates for all strains studied to date are brought together (this study and rates from the literature) there is no clear difference between phytoplankton groups (where there is

sufficient data to make comparisons). For diatoms, rates at ambient levels of iodate (300 – 500 nM) range from -1.65 amol I⁻ cell⁻¹ d⁻¹ in *Thalassiosira pseudonana* (CCMP 1335, Chance et al., 2007) to 123 amol I⁻ cell⁻¹ d⁻¹ in *Nitzschia* sp. (CCMP 580, Chance et al., 2007). In the Prymnesiophytes rates range from 0.7 ± 0.6 amol I⁻ cell⁻¹ d⁻¹ in *Emiliana huxleyi* (RCC 4560, this study) to 95.5 ± 19.5 amol I⁻ cell⁻¹ d⁻¹ in *Calcidiscus leptoporus* (RCC





1164, this study) at ambient iodate. There was no significant difference in iodide production rates between diatoms and prymnesiophytes (Mann-Whitney rank sum test, p > 0.05, n = 20 for diatoms and n = 22 for prymnesiophytes) or between diatoms, prymnesiophytes and phaeocystales (Kruskal-Wallis, p > 0.05, n = 20 for diatoms, n = 15 for prymnesiophytes and n = 6 for phaeocystales) when data from this and previous studies are considered together. These results were the same whether only data from experiments conducted at ambient iodate were included, or data from all experiments (including those at super-ambient iodate levels) were

considered.

### 3.1.2    Iodine to carbon ratios

An alternative way to compare iodide production rates between species and groups is to normalize against activity, such as carbon-fixation rate, rather than cell density. As photosynthetic rate was not measured we use known literature values for cellular carbon (Table 3) to calculate log-phase rates of carbon incorporation into

cellular biomass (equivalent to net primary production, NPP). These rates are then used to calculate the molar ratio of iodate removed or iodide produced (I:C) conversion ratios for each phytoplankton strain used in this study. Ratios are presented in Table 3 and vary between $10^{-6}$ to $10^{-3}$ for I:C. The range of rates found in this study are variable but do encompass the I:C ratios found in field studies, which are on the order of $10^{-4}$ (Chance et al., 2010; Elderfield and Truesdale, 1980; Wong et al., 1976). With our estimated I:C ratios lieing within the

ranges reported from field studies, it can be assumed that the processes that we observe in our monoculture studies are likely transferable to the field. Whilst there is insufficient data to undertake statistical analysis it is clear that, as with the cell normalised rates, the I:C in diatoms and phaeocystales / coccolithophores overlap significantly. Amongst the diatoms, the I:C ratio ranged from 2.0 x $10^{-5}$ (± 1.2 x $10^{-5}$) in *Chaetoceros* sp. (CCMP 1680) to 1.5 x $10^{-4}$ (± 4.4 x $10^{-5}$) in *Chaetoceros* sp. (RCC 4208). In the Prymnesiophytes it was found to range

from 1.5 x $10^{-5}$ (± 5.7 x $10^{-6}$) in *Emiliana huxleyi* (RCC 1210) to 1.1 x $10^{-3}$ (±4.7x$10^{-4}$) in *Phaeocystis* sp. (RCC 1725). The highest I:C amongst the coccolithophores was 4.5 x $10^{-4}$ (± 1.2 x $10^{-4}$) in *Calcidiscus leptoporus* (RCC 1164).

### 3.1.3    Relationship between iodate uptake and iodide production

Fig. 6 shows the relationship between the log phase iodate removal and iodide production rates in 30

phytoplankton cultures from our study and an additional 11 strains from two studies (Chance et al., 2007; Wong et al., 2002), in which the cultures were also supplied with ambient iodate concentrations. Log-phase iodate consumption- and iodide production rates correlate significantly (Fig. 6a; Spearman's Rank, $r_s$ = -0.37, p = 0.018, n = 41) but the correlation for the overall experimental rates is stronger (Fig. 6b; Spearman's Rank, $r_s$ = -0.72, p = 0.000, n = 30). Also shown in the Figure are the "1:1"-lines. Data points below the line suggest higher

iodate removal rates than iodide production, while data points above suggest the opposite. Data points below the line after the end of the experiments (Fig. 6b) indicate loss of iodine during the experiment (or 'missing iodine'). A least squares Regression line on top of the 1:1 line would indicate that all iodate consumed is converted into iodide in the majority of the cultures. The flatter slope of the least squares line (grey line) in Fig. 6a in comparison to that in Fig. 6b suggests higher incorporation of iodate compared to iodide production during the

logarithmic phase. This implies that iodate taken up during active growth is not immediately converted to iodide. Whilst the slope is steeper the least squares regression line in Fig. 6b still does not sit over the 1:1 line





suggesting incomplete conversion of iodate to iodide and 'missing iodine'. The existence of 'missing iodine' is explored further in *Section 3.3*.

### 3.2 Comparison of log and post-log phase rates of iodide production

To investigate if growth stage is an important determinant of the rates of inorganic iodine speciation across diverse phytoplankton groups, we compared logarithmic and post-logarithmic rates of change in iodide (Fig. 7). It is clear from Fig. 7 that there is no general pattern across the strains studied. Some cultures demonstrated higher iodide production rates in the log-phase and others in the post-log phase. A paired t-test revealed that there was no consistent difference between log and post-log phase rates of change in iodide across the
phytoplankton strains included in this study ($p > 0.05$, n = 30).

It is interesting to note that declines in iodide concentrations were observed during the post-log phase in two strains (*Chaetoceros* sp. CCMP1680, $-2.6 \pm 0.5$ amol $I^-$ $cell^{-1}$ $day^{-1}$; *Phaeocystis antarctica* RCC 4024, $-1.1 \pm 1.4$ amol $I^-$ $cell^{-1}$ $day^{-1}$). There is also evidence from the growth curve data that there was a decline in iodide
concentrations during the later stages of the growth curve experiment for *Emiliana huxleyi* (RCC 4560; Fig. 3). It has been established in previous studies that phytoplankton also take up iodide (de la Cuesta and Manley, 2009; Bluhm et al., 2010) and this could explain these declines. Two of the cultures (*Chaetoceros* sp., CCMP 1690; *Emiliana huxleyi*, RCC 4560) had very low iodate (< 10 nM) during the period when iodide concentrations decreased so the cultures may have switched their iodine source to iodide. However one of the cultures where a
decline was observed (*Phaeocystis antarctica*, RCC 4024) still had substantial levels of iodate (~ 180 nM) when iodide decline was observed. In addition to uptake the disappearance of iodide may have also indicated conversion into other organic or inorganic forms. Volatile/low molecular weight organoiodine compounds are usually found in concentrations in the picomolar range both in monocultures (Hughes et al., 2006) and in the field (Hepach et al., 2016). Dissolved organic iodine (DOI) has been suggested to be a possible intermediate step
in the reduction of iodate to iodide. DOI is found in nanomolar ranges in coastal regions with high riverine input but concentrations are lower in open ocean regions (Wong and Cheng, 2001). To date, evaluations of DOI in monocultural batch experiments have not been conducted. However, Wong and Cheng (2001) suggested that DOI could form from microalgal exudates, which could e.g. apply to species such as *Phaeocystis* sp.

### 3.3 Net changes in iodine speciation across experimental duration

The rates of change (normalised to the total experimental duration) and composition of iodine speciation at the end of the experiments in each replicate are shown in Fig. 8. The largest overall net decrease in iodate (mean ± standard deviation, $-313.2 \pm 18.1$ nM) was seen in *Calcidiscus leptoporus*, while the smallest ($-78.6 \pm 26.1$ nM) was seen in *Emiliana huxleyi* (RCC1210). Consistent with this, Fig. 8a shows that the largest overall rate of decline in iodate was observed in *Calcidiscus leptoporus* ($-6.4 \pm 0.4$ nM $d^{-1}$) with the smallest again seen in
*Emiliana huxleyi* (RCC1210; $-1.1 \pm 0.4$ nM $d^{-1}$). The highest net increase in iodide was seen in *Calcidiscus leptoporus* (RCC1164; $272.3 \pm 17.6$ nM) and the lowest was seen in *Synechococcus* sp. in which some changes in inorganic iodine speciation were observed but no significant net increase in iodide was observed across the experiment ($-0.5 \pm 1.3$ nM). Highest overall rates of iodide increase (Fig. 8a) were observed in *Calcidiscus leptoporus* ($5.6 \pm 0.4$ nM $d^{-1}$) and lowest in *Synechococcus* sp. ($0.0 \pm 0.0$ nM $d^{-1}$).




Fig. 8b shows the composition of iodine speciation at the end of each experiment with blue bars indicating 'missing iodine' (difference of net iodate decline and iodide increase). It is apparent that in 23 of the 30 studied culture replicates there is significant 'missing iodine' (i.e. less iodide produced compared to iodate lost from the media). Here 'significant' is defined as more than 10 % of initial iodate given that 10 % is the precision of the measurement (see *Section 2.2.1*). In eight out of 30 replicates, this 'missing iodine' is more than 50 % of the initial iodate concentrations. The 'missing iodine' levels range from $46.9 \pm 33.3$ nM in *Emiliana huxleyi* (RCC 1210) to $257.8 \pm 10.9$ nM in *Chaetoceros* sp. (CCMP 1690). This suggests that there is not always an immediate conversion of iodate to iodide in the medium and that some of the iodate taken up is retained by the cells or converted into (and stored) another form. Previous studies have also observed 'missing iodine' in their phytoplankton cultures (Chance et al., 2007; van Bergeijk et al., 2016; Wong et al., 2002). It is possible that the 'missing iodine' has been converted into organic forms (including volatile organics), other inorganic forms such as hypoiodous acid and molecular iodine, which however are very short-lived in the ocean due to reaction with e.g. organic matter (Luther et al., 1995), or particulate iodine. Establishing the location/form of the 'missing iodine' will require conformation from future studies which include measurements of all forms of iodine (iodate, iodide, particulate iodine, volatile organoiodine compounds, DOI, molecular iodine and hypoiodous acid). Overall observations of 'missing iodine' are not consistent with the mechanism of iodate reduction to iodide proposed by Bluhm et al. (2010) who suggested that iodate discharged during the senescent phase is converted to iodide in the external media following the release of reduced sulfur species upon cell lysis.

It is apparent from Fig. 2 – 5 that although each culture had entered the senescence stage by the end of the experiment, the length of time spent in this stage and the proportion of dead cells present varied between experiments. Given the potential for a link between iodate reduction and cell death (Bluhm et al., 2010) it is important to consider this when exploring differences in the net changes in inorganic iodine speciation across the experimental duration. Fig. 9 presents the ratio of iodide produced to iodate taken up ($I^- : IO_3^-$), average rate of change in iodide across the experimental duration and the net increase in iodide produced across each experiment grouped by senescence stage. Here senescence stage is defined as the % of maximum cells remaining at the end of the experiment and the two groups are late senescence (0 – 50 % cells remaining) and early senescence (51 – 100 % cells remaining). Fig. 9a shows that there is a significant difference in $I^- : IO_3^-$ (Wilcoxon rank sum test, $p = 0.014$, $n = 30$, significance level $p < 0.05$). The average $I^- : IO_3^-$ (Fig. 9a) is significantly higher in cultures at a late stage of senescence (median of ratio = 0.57) compared to those in early senescence (median of ratio = 0.12). This suggests that across a diverse range of phytoplankton cells a greater proportion of the iodate taken up is released as iodide as senescence progresses. This is supported by the Wilcoxon rank sum tests performed on the average rate of change in iodide across senescent stage groups ($p = 0.005$, $n = 30$, Fig. 9b) and total net change in iodide ($p = 0.006$, $n = 30$). The link with cell senescence would not have been apparent from the log/post-log analysis (*Section 3.2*) as this did not consider senescence stage.

Overall our findings suggest that cell death is an important factor controlling iodide production. Considering this and the observation of 'missing iodine' across all phytoplankton groups we propose that: phytoplankton take up or convert iodate to other organic/inorganic forms during active growth; and, that the taken up/converted iodate is reduced to iodide during cell death/senescence. The reduction that occurs upon cell death could be explained by the reduced sulfur mechanism proposed by Bluhm et al. (2010) but there will be a myriad of other chemical





changes that could occur as cells lyse that could also be involved. Iodide production during the active growth
(log) phase can be explained by the low level of cell death that is known to take place even during active growth.
Whilst the environmental stress (e.g. nutrient availability) that occurs in a batch culture over time will clearly

enhance the rate of cell death, natural cell death due to (for example) exhaustion of division potential (age) or
programmed cell death can occur at any time (Franklin et al., 2010).

### 3.4 Implications for process-based models of inorganic iodine cycling in the oceans

The incorporation of phytoplankton functional types (PFTs) into the ecosystem dynamics of ocean
biogeochemical models has led to improved performance and accuracy (Gregg et al., 2003) but our results

suggest this approach would not be suitable for models of inorganic iodine cycling in seawater. Representatives
of common PFTs including pico-autotrophs (e.g. cyanobacteria), phytoplankton silicifiers (e.g. diatoms) and
phytoplankton calcifiers (e.g. coccolithophores; Quéré et al., 2005) have been investigated for iodate to iodide
conversion here and in previous studies. Following the definition in Quéré et al. (2005) each PFT in an inorganic
iodine cycling model would need to have a distinct and explicit role. However in the present study all PFTs

studied to date were found to drive iodate to iodide reduction, there was large variability in rates within PFTs
and we did not find a significant difference in rates of conversion between diatoms and prymnesiophytes, or
diatoms, coccolithophores and phaeocystales. The available evidence suggests that there is no significant
difference in the rates and patterns of iodate to iodide reduction between phytoplankton groups or functional
types.


Our observations of 'missing iodine' and link between iodide production and cell senescence do, however,
provide important guidance for ocean iodine cycling models. These findings suggest that highest iodide
production rates will be observed during the later stages of phytoplankton blooms and there will most likely be a
lag between maximal phytoplankton biomass and the highest iodide concentrations. This suggestion is supported

by time-series measurements in coastal Antarctica (2005 – 2008, Chance et al., 2010) which show that each year
there was a time lag of around 60 days between the onset of the microalgal bloom and the iodide maximum.
These results suggest that the terms for (non-predatory) phytoplankton mortality typically included in
biogeochemical models (e.g. ERSEM, Butenschön et al., 2016) could be used to incorporate iodide production in
to process-based models.

### 4   Conclusions

This study aimed to establish if there are common features of iodate to iodide reduction amongst diverse
phytoplankton that could be used to guide the development of ocean iodine cycling models. By combining our
results with those of previous studies we have shown that there is no significant difference in cell-normalised
iodide production rates between key phytoplankton groups (diatoms versus prymnesiophytes) or phytoplankton

functional types (PFTs, e.g. diatoms versus coccolithophores). We did, however, observe 'missing iodine' in the
majority of phytoplankton cultures studied, and found that the iodide yield is significantly higher in cultures at a
later senescence stage. Resolving the fate of 'missing iodine' may yield useful information on the mechanisms
behind iodate conversion to iodide. Furthermore, in line with a previous time-series study (Chance et al., 2010)
these findings suggest that there will be a lag between maximum iodide production rates and peaks in





phytoplankton biomass/productivity in marine systems. We propose that process-based models of inorganic
iodine cycling could be linked to marine ecosystem models via the phytoplankton mortality term. Future process
studies should focus on whether different environmental and physiological drivers of cell death influence the
iodide yield.

**Author contribution**

CH designed the experiments together with HH and RC. HH conducted the experiments and prepared the
manuscript together with CH. CH provided essential input for the preparation of this manuscript and was a P.I.
of the NERC project 'Iodide in the ocean: distribution and impact on iodine production and ozone loss'. Please
note, HH and CH contributed equally to this manuscript. KH provided methodological knowledge for microalgal
cell counts and provided input during the manuscript preparation. SC contributed to completing the incubation
experiments. RC provided methodological knowledge regarding iodine measurement methods and provided
input during manuscript preparation.

**Acknowledgements**

This study was funded by NERC as part of the project 'Iodide in the ocean: distribution and impact on iodine
production and ozone loss' (NE/N009983/1). We would like to thank Liselotte Tinel for her scientific and
technical input during iodine analysis and Matt Pickering for technical help during the incubation experiments.

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





**Figures**

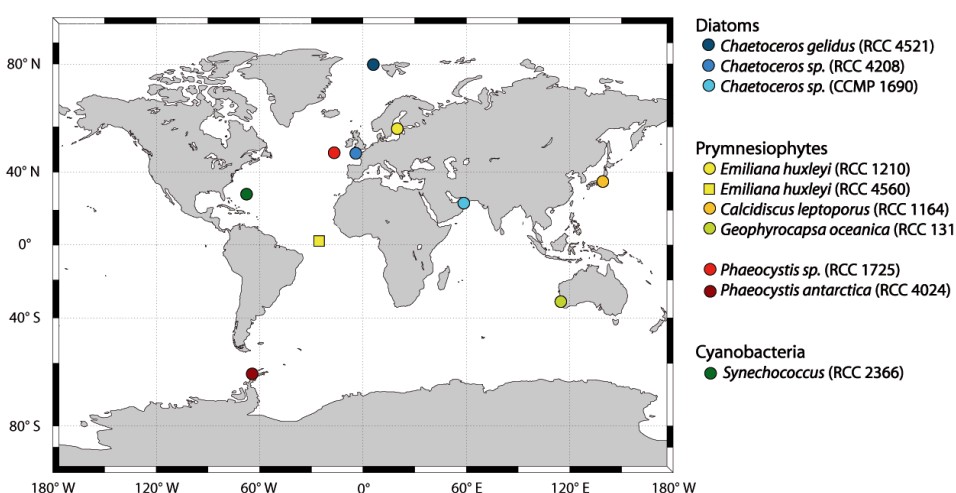

Figure 1. All strains used in the incubation experiments and the original location where they were isolated. Blue colors indicate strains that belong to the group diatoms, yellow to red refer to strains from the class of prymnesiophytes (yellow to orange denotes species from the order of coccolithophores, while red stands for species from the order of phaeocystales), and dark green refers to the only cyanobacteria studied here, *Synechococcus* sp.

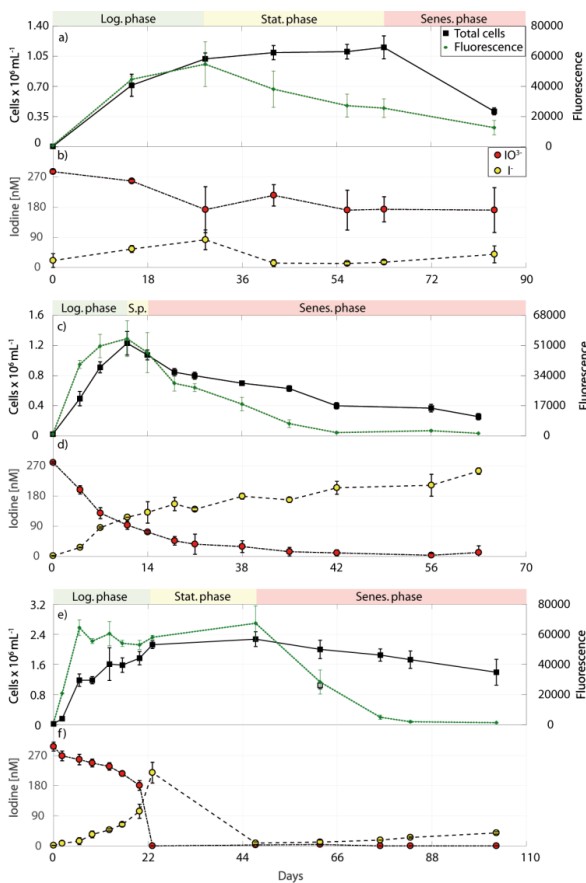

Figure 2. Total cell counts (black – total) and fluorescence readings (green), as well as inorganic iodine speciation (red – iodate, yellow – iodide) over the course of the growth curve experiments of diatoms for *Chaetoceros gelidus* (RCC 4512) in a) and b), *Chaetoceros* sp. (RCC 4208) in c) and d), and *Chaetoceros* sp. (CCMP 1690) in e) and f). The color-shaded bars on top of each graph indicate where the logarithmic phase (green), the stationary phase (yellow), and the senescent phase (red) began and ended based on total cell counts and fluorescence. Values depicted are means from culture replicates with error bars indicating the standard deviations of these means.

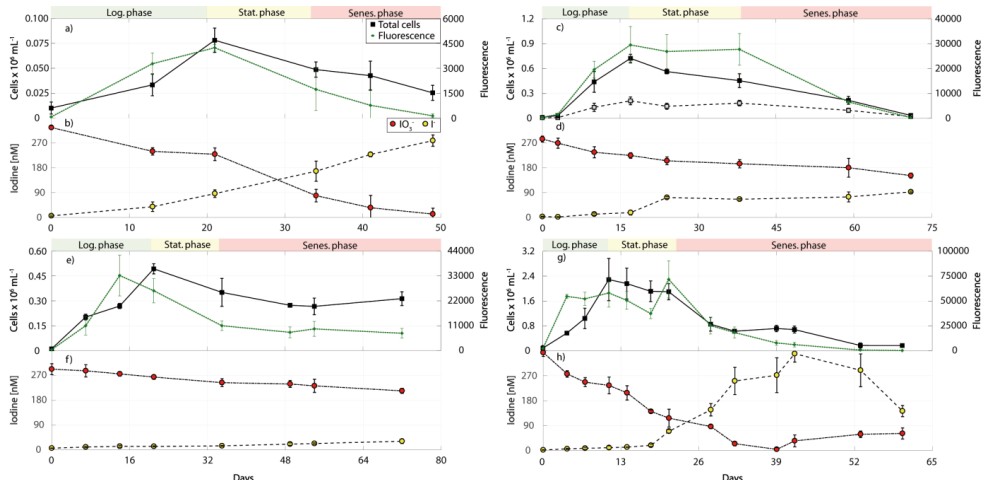


Figure 3. Total cell counts (black – total) and fluorescence readings (green), as well as inorganic iodine speciation (red – iodate, yellow – iodide) over the course of the growth curve experiments of calcifying prymnesiophytes for *Calcidiscus leptoporus* (RCC 1164) in a) and b), *Geophyrocapsa oceanica* (RCC 1318) in c) and d), *Emiliana huxleyi* (RCC 1210) in e) and f), and *Emiliana huxleyi* (RCC 4560) in g) and h). The color-

shaded bars on top of each graph indicate where the logarithmic phase (green), the stationary phase (yellow), and the senescent phase (red) began and ended based on total cell counts. Values depicted are means from culture replicates with error bars indicating the standard deviations of these means.

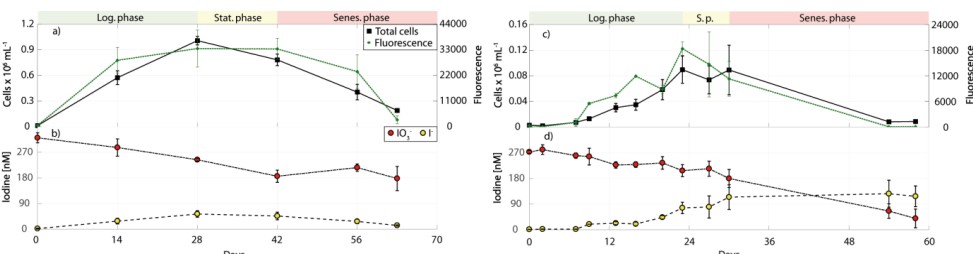

Figure 4. Total cell counts (black – total) and fluorescence readings (green), as well as inorganic iodine speciation (red – iodate, yellow – iodide) over the course of the growth curve experiments of prymnesiophytes for *Phaeocystis antarctica* (RCC 4024) in a) and b), and *Phaeocystis* sp. (RCC 1725) in c) and d). The color-shaded bars on top of each graph indicate where the logarithmic phase (green), the stationary phase (yellow), and the senescent phase (red) began and ended based on total cell counts. Values depicted are means from culture

replicates with error bars indicating the standard deviations of these means.

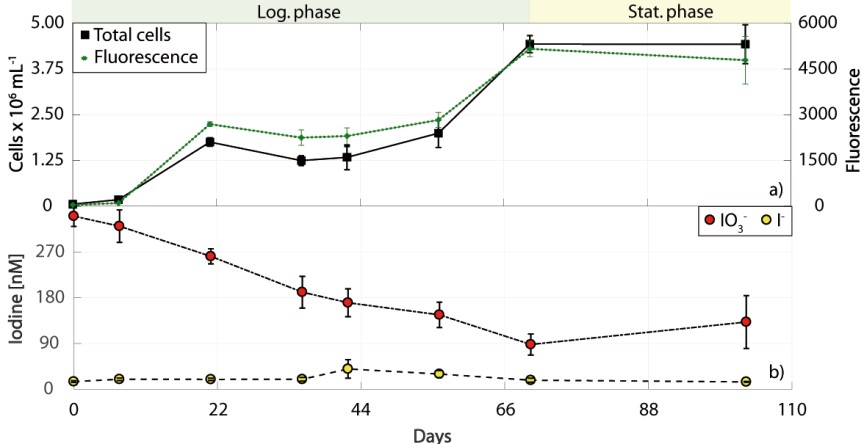

Figure 5. Total cell counts (black – total) and fluorescence readings (green), as well as inorganic iodine speciation (red – iodate, yellow – iodide) over the course of the growth curve experiments of *Synechococcus* sp. (RCC 2366). The color-shaded bars on top of each graph indicate where the logarithmic phase (green), and the stationary phase (yellow) started and ended based on total cell counts. Values depicted are means from culture replicates with error bars indicating the standard deviations of these means.

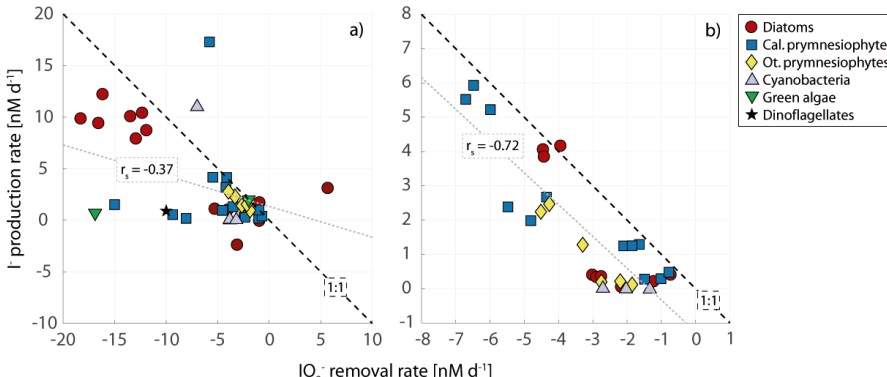

Figure 6. Relationship between a) the log $IO_3^-$ removal and $I^-$ production rate including other studies with similar initial $IO_3^-$ concentrations (Chance et al., 2007; Wong et al., 2002), and b) the overall (whole experiment) removal and production rates in the 30 phytoplankton cultures (3 cultures per strain) from only this study. Rates are calculated as the change in the inorganic iodine species normalized to experimental duration. Dashed line is the 1:1 line. Grey lines are the least square lines (p = 0.018, n = 41; p = 0.00001, n = 30). The marker symbol gives information on whether the shown culture was a diatom (circle), calcifying prymnesiophyte (square), other prymnesiophytes (diamond), cyanobacteria (upward-pointing triangle), green algae (downward-pointing triangle), or dinoflagellate (star). Note that the calcifying prymnesiophyte outlier in a) is from Chance et al. (2007), in which the $I^-$ production rate for *Emiliana huxleyi* (CCMP 373) in the log phase was calculated over the course of less days than the $IO_3^-$ consumption rate due to loss of $I^-$ during the log phase.



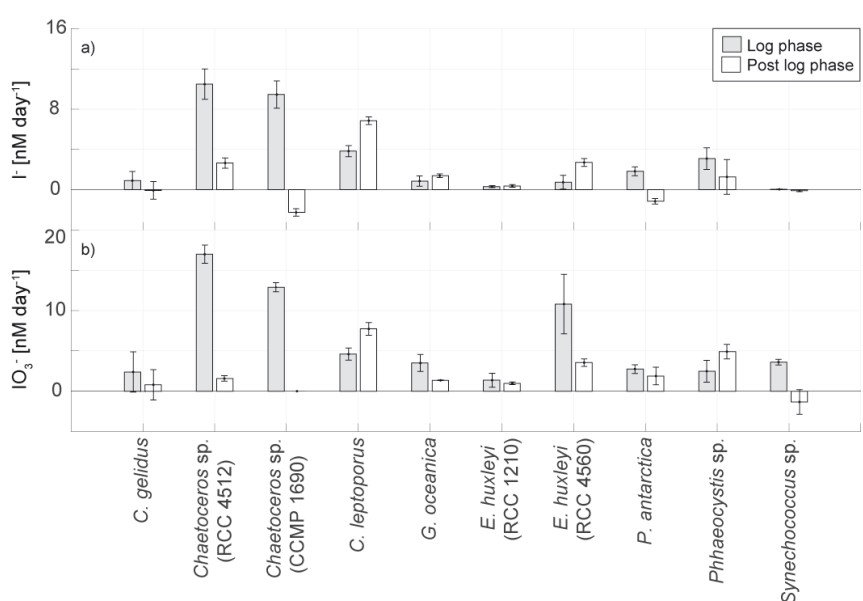


Figure 7. Comparison of the average net change in a) I⁻ and b) IO₃⁻ concentrations during logarithmic (light grey bar) and post-logaithmic (white bar) stages of growth in ten marine phytoplankton cultures averaged over the length of the respective growth phase. Error bars show the standard deviation from three replicate cultures.


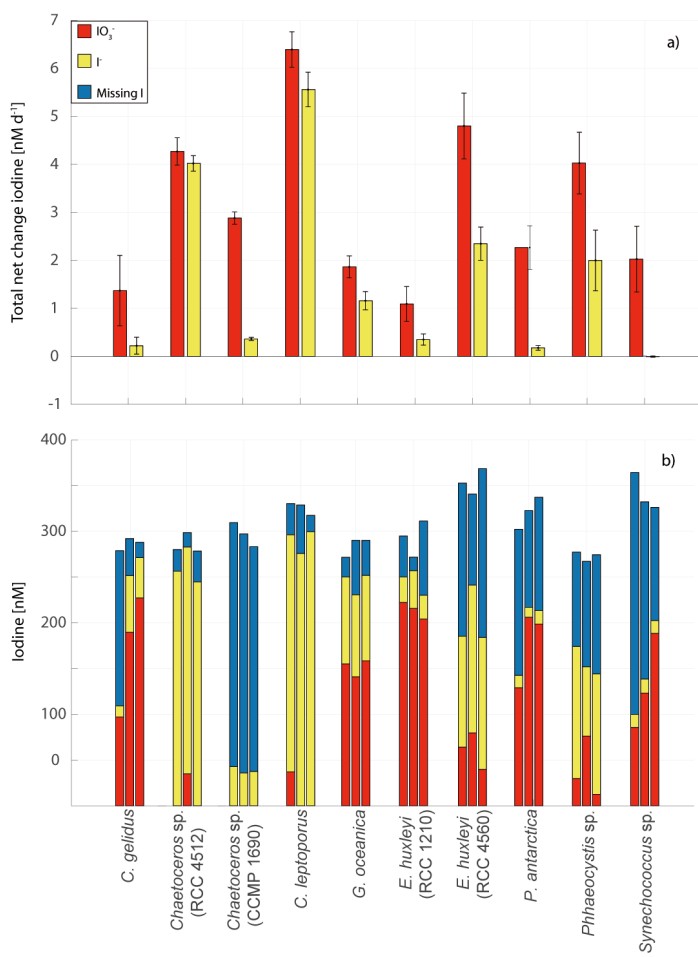

Figure 8. Changes in inorganic iodine speciation in 10 marine phytoplankton cultures: a) overall (whole experiment) rate of change in $IO_3^-$ (red) and $I^-$ (yellow) normalized for the duration of each experiment. Error bars show the standard deviation of three replicate cultures, and b) total $IO_3^-$ (red bar), $I^-$ (yellow bar) and 'missing iodine' (blue bar) for all three replicates for each experiment at the end of each experiment.


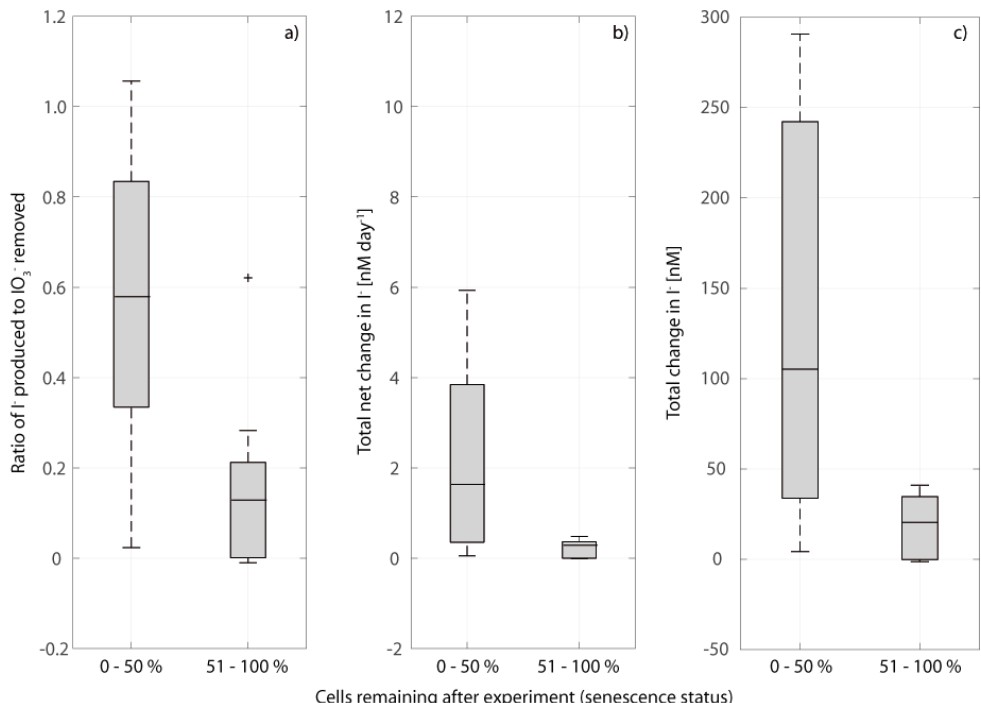

Figure 9. Box plot of the ratio of $I^-$ produced to $IO_3^-$ removed in cultures of a range of marine phytoplankton at
different stages of senescence in a), net rate of change in iodine over the whole length of the experiment (b), and
the total change in iodide at the end of each experiment (c). Senescence status is defined as the % of the
maximum cell number remaining at the end of the experiment: $0 - 50$ % indicates that there are $0 - 50$ % of cells
remaining indicating late senescence; and, a senescence status of $51 - 100$ % indicates that there are $51 - 100$ %
of cells remaining indicating early senescence. Ratios are significantly different between the $0 - 50$ % and $51 -
100$ % groups (Wilcoxon rank sum test, $p = 0.014$, $n = 30$), as is the total net iodine change ($p = 0.005$, $n = 30$)
and the total change in iodide ($p = 0.006$, $n = 30$).

**Tables**


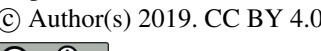



Table 1. Logarithmic-phase cell-normalised rate of iodate removal and iodide production in a range of marine
phytoplankton species investigated in this and previous studies. Experimental conditions are also listed for
comparison. Errors for this study are standard deviations of three replicate cultures.

| Algal group | Species (strain) | $IO_3^-$ (nM) | Temp, $^o$C | Light intensity, $\mu$mol m$^{-2}$ s$^{-1}$ | media | $IO_3^-$ | $I^-$ | Source |
|---|---|---|---|---|---|---|---|---|
| | | | | | | **Rate, amol cell$^{-1}$ d$^{-1}$** | | |
| | | | **Growth conditions** | | | | | |
| Diatoms | *Chaetoceros gelidus* (RCC 4512) | 286 ± 7 | 4 | 50 | K + Si | -4.1 ± 4.6 | 1.5 ± 1.4 | This study |
| | *Chaetoceros* sp. (RCC 4208) | 280 ± 2 | 15 | 75 | K + Si | -26.9 ± 1.8 | 16.6 ± 2.4 | This study |
| | *Chaetoceros* sp. (CCMP 1690) | 297 ± 13 | 25 | 100 | f/2 + Si | -10.8 ± 1.4 | 7.9 ± 0.9 | This study |
| | *Chaeotoceros debilis* (EIFEX) | 5000 | 4 | 50, 100 | f/2+Si | - | 16 ± 3, 14 ± 4 | Bluhm et al. (2010) |
| | *Nitzschia* sp. (CCMP 580) | 300, 10.3 x 10$^3$ | 4 | 60 | f/20 | -148, -6840 | 123, 8600 | Chance et al. (2007) |
| | *Thlassiosira pseudonana* (CCMP 1335) | 300 | 15 | 40 - 50 | f/20 | -2.18 | -1.65 | Chance et al. (2007) |
| | *Pseudo-nitzschia turgiduloides* (EIFEX) | 5000 | 4 | 50, 100 | f/2+Si | - | 493 ± 182, 643 ± 179 | Bluhm et al. (2010) |
| | *Fragilariopsis kerguelensis* (EIFEX) | 5000 | 4 | 50, 100 | f/2+Si | - | 80 ± 17, 93 ± 19 | Bluhm et al. (2010) |
| | *Eucampia antarctica* (CCMP 1452) | 5000 | 4 | 50, 100 | f/2+Si | - | 500 ± 207, 853 ± 124 | Bluhm et al. (2010) |
| | *Phaeodactylum tricornutum* (CCMP1055/15) | 500, 2.5 x 10$^4$, 2.5 x 10$^6$ | 20 | 100 | f/2 | -0.01, -0.4, -38.29 | -, 10.85, 338.3 | van Bergeijk et al. (2016) |
| Prymnesiophytes | *Calcidiscus leptoporus* (RCC 1164) | 326 ± 7 | 20 | 100 | K/2 | -114.9 ± 27.6 | 95.5 ± 19.5 | This study |
| | *Gephyrocapsa oceanica* (RCC 1318) | 284 ± 11 | 17 | 25 | K/2 | -12.0 ± 4.9 | 2.8 ± 1.6 | This study |
| | *Emiliania huxleyi* (RCC 1210) | 293 ± 20 | 17 | 25 | K/2 | -5.6 ± 3.6 | 1.3 ± 0.4 | This study |





| | Species | | | | | | | |
|---|---|---|---|---|---|---|---|---|
| | *Emiliania huxleyi* (RCC 4560) | 354 ± 14 | 20 | 100 | K | -11.0 ± 2.5 | 0.7 ± 0.6 | This study |
| | *Phaeocystis antarctica* (RCC 4024) | 321 ± 18 | 4 | 50 | K/2 | -5.2 ± 1.1 | 3.5 ± 0.9 | This study |
| | *Phaeocystis* sp. (RCC 1725) | 273 ± 5 | 20 | 100 | K/2 | -93.9 ± 25.0 | 60.9 ± 22.5 | This study |
| | *Emiliania huxleyi* (CCMP 373) | 300 | 15 | 40 - 50 | f/20 | -18 | 66.3 | Chance et al. (2007) |
| | *Emiliania huxleyi* (CCMP 371) | 5000 | 18 | 50, 100 | f/2 | - | 11 ± 2, 9 ± 5 | Bluhm et al. (2010) |
| | *Tisochrysis lutea* (CCAP 927/14) | 500, 2.5 x $10^4$, 2.5 x $10^6$ | 20 | 100 | f/2 | -0.2, -2.4, -91.0 | 0.7, 13.8, 195.1 | van Bergeijk et al. (2016) |
| Cyanobacteria | *Synechococcus* sp. (RCC 2366) | 341 ± 20 | 20 | 100 | SN | -2.3 ± 0.3 | 0.0 ± 0.0 | This study |




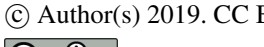



Table 2. Overview of results for the control data for the incubation experiments. Start and end points are shown
       for each measured parameter per type of medium. Values shown are mediums and standard deviations derived
       from all replicates. Media-only-controls were carried out for each incubation set-up with three replicates each,
       and were treated the same way as the inoculated flasks. No significant variations between start and end points
       were detected in any of the parameters shown with respect to detection limits and precision of the methods. Note
that the standard deviations from start and end points is within measurement precision.

| Medium | Time point | $IO_3^-$ [nM] | $I^-$ [nM] |
|--------|-----------|--------------|-----------|
| f/2 + Si | Start | 295.8 ± 6.7 | 3.3 ± 0.7 |
| | End | 304.2 ± 26.2 | 3.7 ± 2.1 |
| K/2 | Start | 236.5 ± 55.4 | 2.7 ± 0.7 |
| | End | 237.4 ± 37.5 | 2.9 ± 0.3 |
| K | Start | 245.6 ± 27.87 | 2.2 ± 0.9 |
| | End | 247.3 ± 13.9 | 9.8 ± 0.4 |
| K + Si | Start | 251.5 ± 38.9 | 2.8 ± 0.7 |
| | End | 249.5 ± 35.5 | 4.6 ± 0.3 |
| SN | Start | 327.9 ± 35.7 | 4.5 ± 2.1 |
| | End | 323.1 ± 27.0 | 11.6 ± 3.8 |





Table 3. Ratios of $IO_3^-$ removal and $I^-$ production to increase in cellular carbon (net primary production, NPP). Also presented are the cellular carbon levels used to make these calculations. Errors are the standard deviations of three replicate cultures.

| | Species (strain) | pgC cell$^{-1}$ | $IO_3^-$:C | Standard deviation | $I^-$:C | Standard deviation |
|---|---|---|---|---|---|---|
| Diatoms | *C. gelidus* (RCC 4512) | 8.2[a] | $4.2 \times 10^{-4}$ | $2.2 \times 10^{-4}$ | $7.1 \times 10^{-5}$ | $6.2 \times 10^{-5}$ |
| | *Chaetoceros* sp. (RCC 4208) | 8.2[a] | $2.3 \times 10^{-4}$ | $2.8 \times 10^{-5}$ | $1.5 \times 10^{-4}$ | $4.4 \times 10^{-5}$ |
| | *Chaetoceros* sp. (CCMP 1690) | 8.2[a] | $8.4 \times 10^{-5}$ | $3.0 \times 10^{-5}$ | $2.0 \times 10^{-5}$ | $1.2 \times 10^{-5}$ |
| Prymnesiophytes | *Calcidiscus leptoporus* (RCC 1164) | 32.5[b] | $5.3 \times 10^{-4}$ | $9.0 \times 10^{-5}$ | $4.5 \times 10^{-4}$ | $1.2 \times 10^{-4}$ |
| | *Gephyrocapsa oceanica* (RCC 1318) | 13.8[c] | $7.4 \times 10^{-5}$ | $2.6 \times 10^{-5}$ | $1.8 \times 10^{-5}$ | $1.0 \times 10^{-5}$ |
| | *Emiliania huxleyi* (RCC 1210) | 10.7[d] | $6.5 \times 10^{-5}$ | $4.0 \times 10^{-5}$ | $1.5 \times 10^{-5}$ | $5.7 \times 10^{-6}$ |
| | *Emiliania huxleyi* (RCC 4560) | 10.7[d] | $6.1 \times 10^{-5}$ | $1.5 \times 10^{-5}$ | $3.8 \times 10^{-6}$ | $3.1 \times 10^{-6}$ |
| | *Phaeocystis antarctica* (RCC 4024) | 9.0[e] | $1.0 \times 10^{-4}$ | $1.6 \times 10^{-5}$ | $6.8 \times 10^{-5}$ | $1.3 \times 10^{-5}$ |
| | *Phaeocystis* sp. (RCC 1725) | 9.0[e] | $8.1 \times 10^{-4}$ | $3.1 \times 10^{-4}$ | $1.1 \times 10^{-3}$ | $4.7 \times 10^{-4}$ |
| Cyanobacteria | *Synechococcus* sp. (RCC 2366) | 0.3[f] | $2.7 \times 10^{-3}$ | $2.2 \times 10^{-4}$ | $2.9 \times 10^{-5}$ | $2.3 \times 10^{-5}$ |

[a]Degerlund et al. (2012); [b]Heinle (2013); [c]Jin et al. (2013), Baumann (2004); [d]Blanco-Ameijeiras et al. (2016); [e]Vogt et al. (2012); [f]Buitenhuis et al. (2012)

