# Peer review of "Senescence as the main driver of iodide release from a diverse range of marine phytoplankton"

_Biogeosciences, 2019_

## Referee Comment (RC1) · Anonymous Referee #1 · 1 Jan 2020

The authors present here a well-written study in which they studied 10 different species of phytoplankton in their ability to reduce iodate to iodide as the reaction of iodide with ozone plays an important role in the depletion of ozone in the atmosphere. It is important to better understand this biological inorganic iodine cycle in the sea surface to be able to use iodide fields in global chemical transport models. The find that in this process iodine is missing and that the stage of the senescence phase plays an important role in this reduction.

Overall the abstract is written in a confusing way in the first half and could use some clarifications, please. Whilst the second half is a lot better and the introduction is well-

written (the first paragraph could do with some chemical equations or an overview figure for the cycle between the marine and the atmospheric parts) and the common thread becomes very clear. In my opinion the title could be improved as it is very broad and doesn't resemble the importance/most important outcome of this study. The authors nicely bring their results into perspective by comparing to the rare previous studies. Some of the figures need to be made easier for the reader and it the size will be crucial in the final paper (not too small). Some of the findings and especially the stages of the senescence phase (Fig. 9) and the missing iodine need significantly more and a thorough discussion to showcase this great dataset better, please for it to be published. Overall, I think it is a good dataset and the topic fits nicely into BGD (GBC would have been a good fit for example as well for example) and the intro and methods section are well-written. The abstract and discussion need to improved for publication. Thanks to the authors for all the work they put into this piece of work.

L13: I don't really understand the wording of iodide fields. Shouldn't first the concentration be mentioned in sea surface waters and then for the models the iodide fields? As first you need the measurements and then you can get to the fields, otherwise it doesn't make sense that you say you need more measurements in the first place, does it? Why does it only depend on sea surface iodide and not iodate as well or even total iodine as you say the iodate can be reduced to iodide and then you might need to take all of this into account? Maybe a good overview graph would be beneficial for the marine to atmosphere reaction, the chemical and the biological pathways and which you measured and why you opted for those to make it easier for the reader. The Teweis et al., 2019, PCCP paper nicely starts with one for example (different, but related). As this is a lot about the inorganic pathway you need to introduce the difference and the importance of the organic cycle in the intro, please, for the reader to understand the differences.

L14-17: Please reword these two sentences. Not an easy to read and understand abstract. I wouldn't continue from there reading this paper as too complicated like: "The

aim of this study was to inform the development of ocean iodine cycling models. . ." To-wards the end of the abstract it gets really well written, understandable and interesting. Thanks.

L31-41: same comment as above, either an overview graph or some chemical reaction equations would be really beneficial here as you can clearly attract significantly more readers which aren't already 100% familiar with this topic.

L57: this paper is on sea surface iodide concentrations, so please use a different reference.

L59: what about the updated version of this paper rather here for the comparisons with more data in it: Chance et al., 2019, Scientific Data?

L73: good paragraph and the last sentence is as what was missing in the abstract in regards to clarity and information why this topic is so important: " Hence we need a greater understanding of biological iodine cycling in order to develop ocean cycling models that can inform studies of ozone deposition to seawater and sea-air iodine emissions."

L81-84: and how do the other groups/species compare and what were their rates like. Please compare better and lead the reader along in this topic. More details necessary.

L115-125: think about introducing the colours for the different groups when introducing the strains.

L126: in regards to the Tsunogai and Sase, 1969 and the possible link to the nitrate reductase and therefore the nitrate concentration wouldn't it make sense to present these here and in Table 1 or better Table2?

L130: after the reference to Bluhm et al., 2010 and the importance of the possible importance of the senescence phase- what does this then mean and wouldn't this have been crucial?

L136: you wrote above that 450-500nM of iodine reflect the natural concentration range in seawater which means that 300-400nM is on the low end of the spectrum. Please reword sentence accordingly.

L149: into which type of bottles?

L166: why in Milli-Q and not in ESAW- what about matrix-differences with this method?

L186: two spaces: cytometry. Samples

Fig2-5 important please see comments below

L211: see comment in regards to the error or rather standard deviation below, please.

L244-266: Please move up your findings in this paragraph and then discuss it with the literature data and not the other way round.

L269: maybe it is simply not possible to group them by phytoplankton groups, but maybe other characteristics, enzymes, . . . possibly the temperature, the environment, . . . are more important?

L299, Fig 6: see important comment below. This needs more of a discussion and context, please.

And L309: isn't this the case as you discussed above because of a very different start concentration, doesn't this influence these plots- please, discuss this in the context of you own discussion further above.

L330: would it make sense to add the iodate/iodide concentrations to this plot to make it easier for the reader to follow the text and the differences?

L332: What about the findings by Smythe-Wright et al., in 2006, GBC and the claim made in this paper about the MeI production? Overall for this paper – what about Prochlorococcus?

L266: Please implement, explain and discuss this statement more- not clear why this

is the case.

L374, Fig 9: great plot.

L385: Please continue the discussion on why the different phases in the senescence phase make such a difference, what this means etc. It is crucial for this paper and its importance and publication to bring this into the bigger picture. The paragraph ends pretty abruptly here and please continue.

L391: Doesn't this contradict L 266?

L411: Please further discuss the missing iodide and use chemical equation and dig into the microbiological literature as what it could be, stored depending on which species,... make suggestions and discuss this further, please. Important finding as above and needs to be expanded.

L3 and 435: Please reconsider the order of your authors. After reading the ms and having understood the substantial amount of time and efforts that went into this lab study, wouldn't it make sense that CH went last as the senior and peer-author of this study? And then that HH was the sole first author even though they contributed equally to the paper as this is what often happens?

L 625, Fig 1: Why does E. hux (RCC 4560) get a quadratic shaped symbol while all the others are dots? Wouldn't it make more sense to use different shapes in case someone prints it out in black and white?

Fig2-5:it would be nice to add the species and their symbol and colour above or into each of the graph to make it easier for the reader to spot the species shown and to compare this to Fig 1 then.

L637: usually with three triplicates you use 3 standard deviations as the analytical error and not one.

Fig 3 and 4 are too small and very hard to read being next to each other. In comparison

Fig 5 has a good size and in the final papers these really shouldn't end up being even smaller, please!

Fig2-5 have you considered the same amount of days in all of the plots and would this make the comparison for the reader easier than as it is now?

Fig6: it is not clear to me how Fig a an b can possibly look so different if a included only two additional other studies. In b the dots seem to be in totally different positions although they were supposed to be included in Fig a as well? If this is only the case because the scales are so different and pretty much a whole range of concentration is excluded then say so in the figure captions, comment, discuss this and maybe mark the square in a which is b pretty much "zoomed" in, please.

Fig7: what does this plot look like if you use the final concentration instead of just the net change? And does it make a difference? Please as commented for figure 2-5 use the colour/symbol coding throughout all your figures.

L725: Please add the concentrations for ESAW.

---

## Referee Comment (RC2) · Anonymous Referee #2 · 5 Jan 2020

**General comments:**

This paper describes changes in iodate and iodide concentrations over the entire growth cycle in cultures of various species of phytoplankton, in order to better understand the purpose and mechanism of iodate to iodide reduction in marine phytoplankton, which would help with the development of process-based models of inorganic iodine cycling in the oceans. It clearly falls within the scope of the journal biogeosciences and it is a clearly written, well-organised manuscript. However, I feel they should have tried to determine what the 'missing iodine' was in this study, since this is an issue that was already discussed in previous papers, and needs to be resolved. Knowing what this missing iodine is will help to achieve a better understanding of the purpose and mechanism of iodate reduction. Also, I do not entirely agree with their conclusion that I--production is a result of cell scenescence. Although this process does seem to occurr, the observation that I- production rate was often higher during the log phase clearly indicates that (an)other mechanism(s) must be at least as important (see specific comments).

**Specific comments:**

l. 220-222, 'Media used...in this nutrient'. Since they did not measure nitrate in the culture media at the end of the experiment, nor C:N ratio in the phytoplankton, they cannot state that nitrogen was not limiting. Moreover, 2.5  $\mu$ M is not a high concentration of nitrate for microalgal cultures and since cultures stopped growing, some element (or light) must have become limiting, although not necessarily nitrogen.

1. 315-316, 'Some cultures...in the post-log phase.' I would say that in 6 of the 10 phytoplankton cultures I-production rate was higher in the log phase than in the post-log phase.

1. 325-326, 'It has been established...Bluhm et al., 2010)' Also Van Bergeijk et al., 2013 (J. Phycol. 49:640-647).

1. 387-393, 'Overall our findings...during active growth.'

In my opinion, I- production mainly as a result of cell scenescence is not evident from Figs. 2-5. Although an increase in I- is seen with a decrease in viable cells at the end of the cultures in Fig. 2b, d, e, 3b and f, in several cases I- concentration was higher at the end of the log phase (Figs 1b, c, 4b, d) than at the end of the scenescent phase, and in most cases, I- production rate was higher during the log phase than during post-log phases. It is highly unlikely thas this was due to the presence of scenescent cells, as they suggest.

The fact that more IO3- was consumed than I- produced could also indicate that IO3- reduced to I- was stored as I- inside the cells, which was only released when cells lysed. I- has been described as an inorganic antioxidant in macroalgae, and although probably present at lower intracellular concentrations in microalgae, it could be used as an intracellular antioxidant during active growth. My point is that although in most cases at the end of the microalgae culture experiments, when cells were lysing, an increasing I- concentration was observed, this clearly was not the only or most important process for I- production.

Please comment.

1. 412-413, 'These findings suggest...highest iodide concentrations.' It would be more correct, based on their findings, that highest iodide concentrations will be observed during later stages of phytoplankton blooms, not production rates.

1. 428-430, 'Furthermore,...in marine systems.' Here also, it would be more correct to say maximum iodide concentrations, instead of production rates.

**Technical corrections:**

1. 39, 'O'Dowd et al., 2002' should be O'Dowd et al., 2010.

1. 71 (and rest of the ms), 'Kupper' should be 'Küpper'.

1. 186, 'less than events 1,000 per second' should be 'less than 1,000 events per second'.

l. 102, 'Javier et al., 2018' should be 'Hernández et al., 2018', and l. 525, 'Javier, L. H.' should be 'Hernández Javier, L.'

288, 'With our estimated I:C ratios lieing...' should be 'With our estimated I:C ratios lying...'
340-341, '...Fig. 8Fehler!...werden.' Delete phrase in German.

---

## Author Comment (AC1) · 3 Feb 2020

The authors present here a well-written study in which they studied 10 different species of phytoplankton in their ability to reduce iodate to iodide as the reaction of iodide with ozone plays an important role in the depletion of ozone in the atmosphere. It is important to better understand this biological inorganic iodine cycle in the sea surface to be able to use iodide fields in global chemical transport models. The find that in

this process iodine is missing and that the stage of the senescence phase plays an important role in this reduction. Overall the abstract is written in a confusing way in the first half and could use some clarifications, please. Whilst the second half is a lot better and the introduction is well-written (the first paragraph could do with some chemical equations or an overview figure for the cycle between the marine and the atmospheric parts) and the common thread becomes very clear. In my opinion the title could be improved as it is very broad and doesn't resemble the importance/most important outcome of this study. The authors nicely bring their results into perspective by comparing to the rare previous studies. Some of the figures need to be made easier for the reader and it the size will be crucial in the final paper (not too small). Some of the findings and especially the stages of the senescence phase (Fig. 9) and the missing iodine need significantly more and a thorough discussion to showcase this great dataset better, please for it to be published. Overall, I think it is a good dataset and the topic fits nicely into BGD (GBC would have been a good fit for example as well for example) and the intro and methods section are well-written. The abstract and discussion need to be improved for publication. Thanks to the authors for all the work they put into this piece of work.

Author: We thank reviewer 1 for this helpful review. We will reply and edit according to the specific comments further below. All changes according to reviewer 1s suggestions will be marked in bold purple. With regards to the title of the paper, we suggest "Senescence as the main driver of iodide release from a diverse range of marine phytoplankton". This is more specific to our findings and draws the readers' attention to the main object of the investigation as we identify it.

L13: I don't really understand the wording of iodide fields. Shouldn't first the concentration be mentioned in sea surface waters and then for the models the iodide fields? As first you need the measurements and then you can get to the fields, otherwise it doesn't make sense that you say you need more measurements in the first place, does it? Why does it only depend on sea surface iodide and not iodate as well or even total

iodine as you say the iodate can be reduced to iodide and then you might need to take all of this into account? Maybe a good overview graph would be beneficial for the marine to atmosphere reaction, the chemical and the biological pathways and which you measured and why you opted for those to make it easier for the reader. The Teiwes et al., 2019, PCCP paper nicely starts with one for example (different, but related). As this is a lot about the inorganic pathway you need to introduce the difference and the importance of the organic cycle in the intro, please, for the reader to understand the differences.

Author: Iodide in the sea surface is the direct factor that is involved in release of molecular iodine and other reactive iodine forms to the troposphere. The iodide in turn of course depends on iodate, which is one of the processes that are involved in deriving iodide fields. Iodide fields themselves reflect iodide concentrations on a more global level. However, due to the fact that measurements are sparse, oceanic modelling is one method to derive iodide fields, which then can be included in atmospheric models to derive the fraction of reactive tropospheric iodine that results from sea surface iodide. We added an overview of reactions that are known into the introduction part but prefer to keep the first half of the abstract in this way, since a more detailed description of these processes would not be beneficial to the length of the abstract itself.

L14-17: Please reword these two sentences. Not an easy to read and understand abstract. I wouldn't continue from there reading this paper as too complicated like: "The aim of this study was to inform the development of ocean iodine cycling models..." Towards the end of the abstract it gets really well written, understandable and interesting. Thanks.

Author: Done.

L31-41: same comment as above, either an overview graph or some chemical reaction equations would be really beneficial here as you can clearly attract significantly more readers which aren't already 100% familiar with this topic.

[Figure]

Author: Done.

L57: this paper is on sea surface iodide concentrations, so please use a different reference.

Author: We updated the reference to Chance et al., 2010.

L59: what about the updated version of this paper rather here for the comparisons with more data in it: Chance et al., 2019, Scientific Data?

Author: Thank you for this suggestion. We prefer citing the 2014-paper since it includes a global estimation and scientific interpretation of the paper.

L73: good paragraph and the last sentence is as what was missing in the abstract in regards to clarity and information why this topic is so important: "Hence we need a greater understanding of biological iodine cycling in order to develop ocean cycling models that can inform studies of ozone deposition to seawater and sea-air iodine emissions."

Author: Thank you for this suggestion, we edited the abstract accordingly.

L81-84: and how do the other groups/species compare and what were their rates like. Please compare better and lead the reader along in this topic. More details necessary.

Author: As we discuss this in more detail in the discussion, we prefer to not put this in too much detail here to avoid repetition. Rates of other studies are also listed in Table 1, we refer to this table as overview.

L115-125: think about introducing the colours for the different groups when introducing the strains.

Author: We have edited this section accordingly.

L126: in regards to the Tsunogai and Sase, 1969 and the possible link to the nitrate reductase and therefore the nitrate concentration wouldn't it make sense to present

these here and in Table 1 or better Table2?

Author: We edited nutrient concentrations of each media into Table 2 as suggested. The phrase in the respective section was edited accordingly as well.

L130: after the reference to Bluhm et al., 2010 and the importance of the possible importance of the senescence phase- what does this then mean and wouldn't this have been crucial?

Author: Cultures in the reference Bluhm et al. (2010) had different "stages" of senescence as well (compare for example Fragilariopsis kerguelensis aand Pseudi-nitzschia turgiduloides in the mentioned reference). However, Bluhm et al., 2010 did not take these different stages of senescence into account when data were interpreted, implying that the specific stage of senescence was not of importance. We on the contrary discuss the importance of the senescence stage within our study, thus adding valuable information to the outcome of the Bluhm-paper.

L136: you wrote above that 450-500nM of iodine reflect the natural concentration range in seawater which means that 300-400nM is on the low end of the spectrum. Please reword sentence accordingly.

Author: Done.

L149: into which type of bottles?

Author: We added this information.

L166: why in Milli-Q and not in ESAW- what about matrix-differences with this method?

Author: The iodate method employed here includes a number of steps which minimise any potential biases effects caused by different sample (saline) and standard (ultrapure water) matrices. Sulfamic acid is first added to create acidic conditions required for iodate reduction, and also to remove nitrite, which is the most significant interferent in the method (Truesdale, 1978). A large excess of potassium iodide (final concentration is

36 mM) is then added in order to form the I3- ion. This excess overwhelms any sample-to-sample or sample-to-standard variation in iodide concentrations, and is also thought to prevent reduction of I3- by redox active material in the sample (Truesdale and Smith, 1979). Iodate concentrations are calculated from the difference in absorbance before and after the addition of potassium iodide reagent, which accounts for differences in other background species which absorb at 350 nm, most notably CDOM. It is possible that salinity differences may have a slight impact on the reaction rate, but these are expected to be insignificant given the very large iodide excess and the 2.5 minute reaction time allowed between the two readings. Salinity differences could potentially also affect the method through their impact on optical properties, including the refractive index of the solutions. We do not believe this effect to be substantial in the method as used, as we achieve quantitative recovery of the ∼300 nM spike added to the culture media solutions (Table 2). However, we thank the reviewer for alerting us to this issue as we acknowledge that in future work the use of saline standards may further improve the method performance.

L186: two spaces: cytometry. Samples Fig2-5 important please see comments below

Author: Done.

L211: see comment in regards to the error or rather standard deviation below, please.

Author: See comment further down.

L244-266: Please move up your findings in this paragraph and then discuss it with the literature data and not the other way round.

Author: Done.

L269: maybe it is simply not possible to group them by phytoplankton groups, but maybe other characteristics, enzymes, ... possibly the temperature, the environment,...are more important?

Author: We have done statistical analysis that also includes region, temperature and so

on. However, we did not find any similarities. Part of this may be due to the low number of strains from different regions, which makes grouping these species in a statistical meaningful way difficult. Thus, we prefer keeping this discussion as it is.

L299, Fig 6: see important comment below. This needs more of a discussion and context, please.

Author: See also comment below.

And L309: isn't this the case as you discussed above because of a very different start concentration, doesn't this influence these plots- please, discuss this in the context of you own discussion further above.

Author: Only culture studies where iodate was added to the medium in ambient concentrations were included in this analysis, which we mentioned in the respective section and the description of the figure. Large differences in iodide production from iodate could be observed when iodate was added in large excess. We assume that processes that take place in similar concentration ranges, as was the case here, are comparable. Thus, we prefer leaving the discussion as it is.

L330: would it make sense to add the iodate/iodide concentrations to this plot to make it easier for the reader to follow the text and the differences?

Author: We believe adding the concentrations to Fig. 7 would make this figure unclear/confusing. We however add in a reference to the respective figures for the development of iodide/iodate concentrations during the course of the experiment (where logarithmic phase and so on is marked) to clarify this discussion.

L332: What about the findings by Smythe-Wright et al., in 2006, GBC and the claim made in this paper about the MeI production? Overall for this paper – what about Prochlorococcus?

Author: Although Smythe-Wright et al., 2006 did observe methyl iodide ($CH_3I$) in high concentrations, this was still in the picomolar range, which is compared to inorganic

iodine forms one magnitude lower. We agree that Prochlorococcus sp. may be a key species given its' wide-spread occurrence. However, since this species was not part of the presented investigations here and there is also great uncertainty with respect to the ability of Prochlorococcus sp. to be involved in CH3I production (e.g. Brownell et al., 2010 who found very low production rates). Hughes et al., 2011 suggested that the production of CH3I from this species may largely depend on the physiological state of the cells themselves. Given these uncertainties, we prefer to not include this species in the discussion in this paper here.

L366: Please implement, explain and discuss this statement more- not clear why this is the case.

Author: The process as proposed by Bluhm et al., 2010 suggests that iodate, which is present in the medium is directly converted into iodide when reduced sulfur species are released from the cells during cell lysis, which leads to the appearance of iodide in the senescent phase (suggesting an extracellular, rather direct process rather than intracellular conversion). However, 'missing iodine' essentially means that iodate is decreasing in a higher magnitude than iodide appears, which means that either iodate is converted into another form of iodine than investigated in our study or the iodate conversion into iodide is an intracellular process (meaning that iodate is in the particulate fraction), which takes some time (or both processes could play a role). Additionally, one issue of the study of Bluhm et al., 2010 is that iodate was added into the medium in large excess (5 $\mu$M) and that the method used to determine iodate was not precise enough to observe loss of iodate within the study. Hence, Bluhm et al., 2010 were not able to observe the phenomenon of 'missing iodine', which was consequently not included in their discussion of possible processes involved in iodate conversion. Iodide could however certainly be directly released in the senescent phase due to cell lysis, which could explain why 'missing iodine' is more strongly observed in cultures that were in an earlier stage of senescence. We include a more detailed explanation in this section.

L374, Fig 9: great plot.

Author: Thank you!

L385: Please continue the discussion on why the different phases in the senescence phase make such a difference, what this means etc. It is crucial for this paper and its importance and publication to bring this into the bigger picture. The paragraph ends pretty abruptly here and please continue.

Author: We added a small paragraph.

L391: Doesn't this contradict L 366?

Author: Not necessarily. We cannot completely rule out the process mentioned by Bluhm et al., 2010. However, the phenomenon of 'missing iodide' at least suggests that direct conversion from iodate to iodide may not be the main process, or the reduction of iodate to iodide involves more reactions in-between. We edit this section clearer, so that it does not appear as a contradiction.

L411: Please further discuss the missing iodide and use chemical equation and dig into the microbiological literature as what it could be, stored depending on which species,...make suggestions and discuss this further, please. Important finding as above and needs to be expanded.

Author: We agree that this 'missing iodine' may be an important factor in disentangling the processes underlying iodate conversion to iodide. Unfortunately, we do not believe that our experimental set up allows for drawing more detailed conclusions and we believe that doing this would be too speculative. We however strongly recommend to doing more studies on a metabolic level and including all potential species of iodine in the measurements.

L3 and 435: Please reconsider the order of your authors. After reading the ms and having understood the substantial amount of time and efforts that went into this lab study, wouldn't it make sense that CH went last as the senior and peer-author of this

study? And then that HH was the sole first author even though they contributed equally to the paper as this is what often happens?

Author: Since the input and time that CH spent on this study exceeds this of senior/peer-authors that appear last on scientific papers, we prefer keeping the order of authors as it appears now.

L625, Fig 1: Why does E. hux (RCC 4560) get a quadratic shaped symbol while all the others are dots? Wouldn't it make more sense to use different shapes in case someone prints it out in black and white?Fig2-5:it would be nice to add the species and their symbol and colour above or into each of the graph to make it easier for the reader to spot the species shown and to compare this to Fig 1 then.

Author: This was done because we have two different strains of the same species. That's why we chose different symbols. But we agree that it may be confusing. The Figure is edited accordingly with only circles as symbols.

L637: usually with three triplicates you use 3 standard deviations as the analytical error and not one.

Author: The standard error here does not refer to the analytical measurement error from the voltammetry but to the experimental error due to the procedure/set up of the cultures. Our error is supposed to represent the precision of the experimental set up and the ability of the culture to produce iodide from iodate. We clarify this in the description of the standard deviation that we use here.

Fig 3 and 4 are too small and very hard to read being next to each other. In comparison Fig 5 has a good size and in the final papers these really shouldn't end up being even smaller, please!

Author: Fig. 3 and 4 are arranged now in the same manner as Fig. 2.

Fig2-5 have you considered the same amount of days in all of the plots and would this make the comparison for the reader easier than as it is now?

Author: We agree that this would make the plots more comparable. However, some of the investigated species were very long-living such as the Arctic species Chaetoceros gelidus (RCC 4512) that we incubated for almost 90 days. Other species such as Calcidiscus leptoporus (RCC 1164) reached the senescent phase much quicker and experiments could thus be stopped earlier. The large logistical effort to investigate the amount of species as we did constrained us to this experimental set-up. Showing less days for the cultures would take away important information for the cultures that had to be grown for a longer time. Showing more days for all of the cultures would decrease the readability of the cultures that we investigated for a shorter time period. We thus decided to leave the figures with this time period (this was similarly done e.g. by Bluhm et al., 2010).

Fig 6: it is not clear to me how Fig a and b can possibly look so different if a included only two additional other studies. In b the dots seem to be in totally different positions although they were supposed to be included in Fig a as well? If this is only the case because the scales are so different and pretty much a whole range of concentration is excluded then say so in the figure captions, comment, discuss this and maybe mark the square in a which is b pretty much "zoomed" in, please.

Author: The two figures look different because different stages of the experiment are shown here. The first shows only rates during the logarithmic part of the experiment. The second part of the figure shows production vs incorporation at the end of our experiment. We could not find any studies that investigated iodide production from iodate over all growth stages at ambient iodate concentrations, which is why we could not include them in the second part of the figure. If the regression line would be on top of the '1:1-line', this would indicate that all iodate that is taken up results into iodide. However, the regression line is much flatter in the logarithmic phase than at the end of the experiment, which indicates that there is a clear time gap between iodate take-up and iodide release. The difference between the two parts of the figure is more clearly described now.

Fig7: what does this plot look like if you use the final concentration instead of just the net change? And does it make a difference? Please as commented for figure 2-5 use the colour/symbol coding throughout all your figures.

Author: We agree with the reviewer that similar colours etc. are very helpful for the reader. However, Fig. 7 shows different parameters than Fig. 8 – 9, so we believe that similar coding would be misleading here (colour coding in Fig. 8 for example is supposed to represent different species of iodine, while these are shown on different axes in Fig. 7). The suggested final concentrations are shown in Fig. 8b), which does look different to the figure shown here. However, our intention with Fig. 7 was to show and further investigate the time gap between incorporation of iodate and production of iodide, which is more apparent when investigating the net change rather than the total concentrations (this also accounts for slightly different starting concentrations). Thus we believe that the net change is s suited parameter here.

L725: Please add the concentrations for ESAW.

Author: We added the concentrations of the nutrients to one of the tables. The concentrations of artificial seawater, i.e. ESAW, are well known and applied. The according reference is mentioned in the respective section (2.2). All ingredients used did not contain iodine compounds. We would thus refer to Berges et al., 2001 for exact concentrations of the components.

References: Berges, J. A., Franklin, D. J., and Harrison, P. J.: Evolution of an artificial seawater medium: Improvements in enriched seawater, artificial water over the last two decades, J. Phycol., 37, 1138-1145, 10.1046/j.1529-8817.2001.01052.x, 2001. Bluhm, K., Croot, P., Wuttig, K., and Lochte, K.: Transformation of iodate to iodide in marine phytoplankton driven by cell senescence, Aquat. Biol., 11, 1-15, 10.3354/ab00284, 2010. Brownell, D., Moore, R., and Cullen, J.: Production of methylhalides byProchlorococcus and Synechococcus, Global Biogeochem. Cy., 24, GB2002, doi: 10.1029/2009GB003671, 2010. Hughes, C., Franklin, D., and Malin,

G.: Iodomethane productionby two important marine cyanobacteria: Prochlorococcus marinus (CCMP 2389) and Synechococcus sp. (CCMP 2370), Mar.Chem., 125, 19–25, 2011. Smythe-Wright, D., Boswell, S., Breithaupt, P., Davidson, R., Dimmer, C., and Eiras Diaz, L.: Methyl iodide production in the ocean: Implications for climate change, Global Biogeochem. Cy., 20, GB3003, doi:10.1029/2005GB002642, 2006. Truesdale, V. W.: The automatic determination of iodate- and total-iodine in seawater, Mar. Chem., 6, 253-273, https://doi.org/10.1016/0304-4203(78)90034-8, 1978. Truesdale, V. W., and Smith, C. J.: A comparative study of three methods for the determination of iodate in seawater, Mar. Chem., 7, 133-139, https://doi.org/10.1016/0304-4203(79)90005-7, 1979.

---

## Author Comment (AC2) · 3 Feb 2020

General comments: This paper describes changes in iodate and iodide concentrations over the entire growth cycle in cultures of various species of phytoplankton, in order to better understand the purpose and mechanism of iodate to iodide reduction in marine phytoplankton, which would help with the development of process-based models of inorganic iodine cycling in the oceans.It clearly falls within the scope of the journal

biogeosciences and it is a clearly written, well-organised manuscript. However, I feel they should have tried to determine what the 'missing iodine' was in this study, since this is an issue that was already discussed in previous papers, and needs to be resolved. Knowing what this missing iodine is will help to achieve a better understanding of the purpose and mechanism of iodate reduction. Also, I do not entirely agree with their conclusion that I–production is a result of cell scenescence. Although this process does seem to occurr, the observation that I–production rate was often higher during the log phase clearly indicates that (an)other mechanism(s) must be at least as important (see specific comments).

Author: We thank the reviewer for this helpful review. We agree that finding the mechanism behind the 'missing iodide' may be the key to determining the processes behind iodate reduction and iodide production, especially with respect to potential functions of this process. However, we do not feel like we can resolve the 'missing iodine' any further on the basis of our experiments. We therefore strongly advice for future studies to further estimate this. We agree that cell senescence per se is not the (only) driver for iodide production, since we could see release of iodide during all stages. Senescence however does play a significant role with respect to the total iodine budget added in the beginning of the experiments as our statistical analysis shows. 'Missing iodine' decreases strongly when algal cultures reached a later stage of senescence (or the iodine budget is more balanced with progressing stage of senescence, respectively), which hints towards a release/production in the latter growth stages. This could potentially be explained with storage of iodine, either in the form of iodate or iodide, within the algal cells (which is then transformed or released later on). The latter interpretation, release after storage, is added into the respective section. Changes in the manuscript according to suggestions from Reviewer 2 will be marked in bold green.

Specific comments: l220-222, 'Media used...in this nutrient'. Since they did not measure nitrate in the culture media at the end of the experiment, nor C:N ratio in the phytoplankton, they cannot state that nitrogen was not limiting. Moreover, 2.5 $\mu$M is

not a high concentration of nitrate for microalgal cultures and since cultures stopped growing, some element (or light) must have become limiting, although not necessarily nitrogen.

Author: It is true that we did not measure nitrate. We also agree that some factor must have become limiting. With this section, we wanted to point out that nitrate was not added in low concentrations to the medium, especially with regard to nitrate values generally found in oceanic regions (Bristow et al., 2017). As the reviewer also pointed out, many cultures released iodide also during the log phase, where nitrate was surely not limiting yet. Thus, we still believe that nitrate was not involved in the process that led to iodide production from iodate. We edited accordingly.

l315-316, 'Some cultures...in the post-log phase.' I would say that in 6 of the 10 phytoplankton cultures I–production rate was higher in the log phase than in the post-log phase. l. 325-326, 'It has been established...Bluhm et al., 2010)' Also Van Bergeijk et al., 2013 (J. Phycol. 49:640-647).

Author: Of the 30 cultures we investigated, 14 had the highest release in the log phase, while 16 released most iodide post-log. We added the actual numbers and we include the reference.

l387-393, 'Overall our findings...during active growth.' In my opinion, I- production mainly as a result of cell scenescence is not evident from Figs. 2-5. Although an increase in I- is seen with a decrease in viable cells at the end of the cultures in Fig. 2b, d, e, 3b and f, in several cases I- concentration was higher at the end of the log phase (Figs 1b, c, 4b, d) than at the end of the scenescent phase, and in most cases, I- production rate was higher during the log phase than during post-log phases. It is highly unlikely thas this was due to the presence of scenescent cells, as they suggest. The fact that more IO3- was consumed than I- produced could also indicate that IO3- reduced to I-was stored as I- inside the cells, which was only released when cells lysed. I- has been described as an inorganic antioxidant in macroalgae, and although probably present at lower intracellular concentrations in microalgae, it could be used as an intracellular antioxidant during active growth. My point is that although in most cases at the end of the microalgae culture experiments, when cells were lysing, an increasing I- concentration was observed, this clearly was not the only or most important process for I- production. Please comment.

Author: We agree (also see comment further up) that cell senescence may not be crucial for iodide reduction itself but it seems from our experiments that it plays a role in iodide release (in comparison to iodate added into the medium, 'missing iodine'). Storage of iodate or iodide (or another form of iodine) may play a significant role. Thus, the phenomenon 'missing iodine' is one key factor to untangle the processes behind iodate reduction to iodide production and what exactly triggers the release or transformation to iodide, respectively. As mentioned above, this is added in now.

l412-413, 'These findings suggest...highest iodide concentrations.' It would be more correct, based on their findings, that highest iodide concentrations will be observed during later stages of phytoplankton blooms, not production rates.

Author: We agree that iodide could be stores within the cells. Thus, we corrected to "iodide release rates".

l428-430, 'Furthermore,...in marine systems.' Here also, it would be more correct to say maximum iodide concentrations, instead of production rates.

Author: We again corrected to "iodide release rates".

Technical corrections: l. 39, 'O'Dowd et al., 2002' should be O'Dowd et al., 2010

Author: Actually, the 2002-citation directly addresses iodine involvement in aerosol formation/new particle formation. Thus we prefer leaving the 2002-paper in as citation.

l. 71 (and rest of the ms), 'Kupper' should be 'Küpper'.

Author: Done.

l. 102, 'Javier et al., 2018' should be 'Hernández et al., 2018', and l. 525, 'Javier, L. H.' should be 'Hernández Javier, L.'

Author: Done.

l. 186, 'less than events 1,000 per second' should be 'less than 1,000 events per second'.

Author: Done.

l. 288, 'With our estimated I:C ratios lieing...' should be 'With our estimated I:C ratios lying...'

Author: Done.

l. 340-341, '...Fig. 8Fehler!...werden.' Delete phrase in German.

Author: Done.

References: Bristow, L. A., Mohr, W., Ahmerkamp, S., and Kuypers, M. M. M.: Nutrients that limit growth in the ocean, Curr. Biol., 27, R474-R478, 10.1016/j.cub.2017.03.030, 2017.

---

## Author Response (AR2)

**bg-2019-443: Senescence as the main driver of iodide release from a diverse range of marine phytoplankton**

Helmke Hepach[1*+], Claire Hughes[1+], Karen Hogg[2], Susannah Collings[1] and Rosie Chance[3]

5    Reply to editor:

Dear editor,

Thank you very much. We incorporated all requested edits. Thank you for accepting our manuscript!